# ROS-Based Nanoparticles for Atherosclerosis Treatment

**DOI:** 10.3390/ma14226921

**Published:** 2021-11-16

**Authors:** Xin Hu, Pengxuan Zhao, Yongping Lu, Yani Liu

**Affiliations:** 1Department of Medical Ultrasound, Tongji Hospital, Tongji Medical College, Huazhong University of Science and Technology, Wuhan 430030, China; xinhu1212@163.com (X.H.); zhaopengxuan@tjh.tjmu.edu.cn (P.Z.); 2Department of Ultrasound, The Affiliated Hospital of Yunnan University, Kunming 650021, China; luyongp@163.com

**Keywords:** reactive oxygen species (ROS), atherosclerosis, nanoparticles, scavenger, ROS-responsive therapy

## Abstract

Atherosclerosis (AS), a chronic arterial disease, is the leading cause of death in western developed countries. Considering its long-term asymptomatic progression and serious complications, the early prevention and effective treatment of AS are particularly important. The unique characteristics of nanoparticles (NPs) make them attractive in novel therapeutic and diagnostic applications, providing new options for the treatment of AS. With the assistance of reactive oxygen species (ROS)-based NPs, drugs can reach specific lesion areas, prolong the therapeutic effect, achieve targeted controlled release and reduce adverse side effects. In this article, we reviewed the mechanism of AS and the generation and removal strategy of ROS. We further discussed ROS-based NPs, and summarized their biomedical applications in scavenger and drug delivery. Furthermore, we highlighted the recent advances, challenges and future perspectives of ROS-based NPs for treating AS.

## 1. Introduction

Atherosclerosis (AS) is the leading cause of death in western developed countries [1,2,3,4]. Currently, clinical treatment strategies focus on controlling risk factors, relieving symptoms and preventing future cardiac events. To date, the most common strategy is drug therapy. For example, the widely used statins could contribute to the inhibition of the formation and progression of AS [5]. In addition to drugs, further stent-assisted therapies or coronary artery bypass surgery have been used for advanced AS to prevent adverse cardiac events [6]. In order to prevent and treat AS and reduce the occurrence of cardiovascular and cerebrovascular events more efficiently, some new strategies, including anti-inflammatory therapy and immunotherapy for AS, are also being extensively studied [7,8]. The typical features of atherosclerotic lesions are the abnormal accumulation of lipids and progressive inflammation in vascular endothelial cells. Moreover, various stimuli, including low-density lipoprotein (LDL), oxidative active substances, infection, mechanical stress and chemical damage, could speed up the process of AS [9]. At present, oxidative stress is considered the main mechanism of AS, and it is widely believed that oxidative stress is an imbalance between the antioxidant capacity and activity species [10,11]. Activity species include reactive oxygen species (ROS), nitrogen and halogen species. Increasing evidence has demonstrated that ROS play a crucial role in the occurrence and progression of AS [12].

ROS, which participate in a series of physiological and pathological processes in the body, are the byproducts of aerobic metabolism. The ROS family is composed of free radicals and non-free radicals. Free radicals mainly include hydrogen peroxide (H_2_O_2_), singlet oxygen (1O_2_), superoxide ion (O_2_•^−^) and hydroxyl radical (OH•^−^). Non-free radicals include peroxynitrite (ONOO-) and hypochlorous acid (HOCl) [13,14]. ROS-dependent modification is the basis for the transduction of intracellular signals that control pleiotropic functions. Nevertheless, excessive ROS may induce a battery of inflammatory responses and exacerbate oxidative stress in cells and tissues. More importantly, the accumulation of ROS inducts lipid peroxidation and glycoxidation reactions, leading to protein cross-linking and aggregation, which causes cell damage and death.

Elevating ROS has been proven in the literature to be an effective strategy for treating tumors [15]. In recent years, some researchers have increasingly focused on ROS scavenging strategies, which could effectively inhibit the formation of foam cells and significantly improve the stability of atherosclerotic plaque. With ROS-based NPs, drugs could reach specific lesion areas, prolong the therapeutic effect, achieve targeted controlled release and reduce adverse side effects. In this review, we reviewed the mechanism of AS and the generation and removal strategy of ROS. We further discussed the ROS-based NPs, and summarized their biomedical applications in scavenger and drug delivery. Furthermore, we highlighted the recent advances, challenges and future perspectives of ROS-based NPs for treating AS.

Existing literature was reviewed using Pubmed and Google Scholar. The following keywords were used to search for literature in Pubmed and Google Scholar: “mechanism and atherosclerosis”, “treatment and atherosclerosis”, “reactive oxygen species”, “nanoparticles”, “reactive oxygen species, nanoparticles, atherosclerosis and treatment”. The inclusion criteria for references were as follows: (1) The NPs are used as drug delivery vehicles for anti-AS therapy or as therapeutics; (2) the NPs are capable of responding to ROS or reducing ROS; (3) the articles are highly credible and influential. Finally, 88 studies were included in this literature review.

## 2. Reactive Oxygen Species and Atherosclerosis

### 2.1. Mechanism of Atherosclerosis

AS is a chronic inflammatory disease of the arterial intima, in which the infiltration and accumulation of lipids and inflammatory cells are the internal driving forces for the progression of the lesions. These inflammatory cells, especially macrophages, mast cells and T lymphocytes and release many cytokines, which promote ROS generation to stimulate the migration of smooth muscle cells and the deposition of collagen, leading to the development of an atheromatous plaque [16]. In addition, inflammation is triggered by an immune response in which T helper type 1 (Th 1) cells are involved in the formation of atherosclerotic plaques [17]. For many years, it has been believed that hypertension, diabetes, hyperlipidemia, obesity, and smoking are risk factors for AS [18]. In general, atherosclerotic lesions are the final result of multiple pathogenic factors [19]. Although current lipid-lowering strategies could effectively slow the progression of AS, the risk of cardiovascular events remains high. In the past few decades, researchers have identified that an excess of ROS could promote the progression of AS [20]. This opens up a new approach for the treatment of atherosclerotic lesions.

### 2.2. Generation of ROS

ROS are necessary mechanisms for organism growth, health and aging, and are important in adjusting various physiological activities. However, excessive production of ROS will cause mitochondrial deoxyribonucleic acid (DNA) and nuclear DNA damage and activate various signaling pathways, such as mitogen-activated protein kinase (MAPK), nuclear factor kB (NF-κB) and janus kinase/signal transducers and activators of transcription (JAK/STAT), which induces a cascade reaction to cell apoptosis [21,22]. Therefore, it is critical to understand excess ROS damage and the source of ROS. ROS come from various cells (such as foam cells, vascular smooth muscle cells, endothelial cells), organelles (such as mitochondria, peroxisomes and endoplasmic reticulum) and cytoplasm. Among them, the ROS produced by mitochondria have the highest content [23,24,25]. In addition, ROS can be produced by enzymatic and non-enzymatic pathways. The enzyme sources of ROS include NADPH oxidase (NOX), lipoxygenase (LOX), cyclooxygenase (COX), xanthine oxidase (XO), uncoupled nitric oxide synthase (NOS), myeloperoxidase and many other amine oxidases [26,27,28,29,30]. Additionally, some risk factors in daily life and work, such as bad living habits and exposure to harmful radiation, can increase the expression of ROS, which leads to more serious inflammatory damage. However, some researchers have indicated that the generation of a large amount of ROS can promote the autophagy of macrophages to slow down the process of AS [31,32].

## 3. Strategies to Reduce ROS for Atherosclerosis Treating

The dynamic balancing of ROS production and clearance is essential for oxidation-reduction equilibrium and cardiovascular fitness. Generally, the elimination of ROS depends on enzymatic and non-enzymatic pathways [13,19]. Nicotinamide adenine dinucleotide (NADH) transforms ROS into water via mitochondrial intima. Certain enzymes in the body, including superoxide dismutase (SOD), catalase (CAT) and peptide peroxidase (GPx), have the ability to clear ROS [19,22,33]. O_2_•^−^ is catalyzed into hydrogen peroxide (H_2_O_2_) by SOD [34]. Next, CAT, GPx and glutathione (GSH) can efficiently convert H_2_O_2_ into nontoxic products, avoiding the accumulation of ROS [35,36]. Due to their poor stability and mass production difficulties, the application of natural enzymes is limited. Therefore, it is very important to develop nanomaterials with enzyme simulation properties to achieve ROS regulation. These nanomaterials have significant advantages in ROS scavenging. Compared with natural enzymes, they have a broad spectrum of ROS scavenging abilities, strong stability in physiological environments, and satisfying biocompatibility and biosafety.

In addition, various type of antioxidants, such as polyphenols, vitamin E and C, flavonoids, ferulic, tempol, statins, probucol and its derivatives, immunosuppressants and glucocorticoid play a crucial role in the prevention and treatment of AS through different mechanisms [37,38,39,40,41,42,43,44,45,46]. However, systemic exposure, off-target effects and poor bioavailability remain concerns for drug therapy [47,48]. Meanwhile, excessive antioxidants may also promote the occurrence and development of AS [49]. It is worth considering that the dosage and delivery of antioxidants used to treat AS are important factors.

## 4. ROS-Based Nanoparticles for Atherosclerosis Treatment

Nanoparticles (NPs) have the characteristics of a small size, large surface and high possibility of biological modification. Not only can they evade the mononuclear macrophage system and prolong blood circulation, but they can also effectively target lesions [15,50]. Targeting of AS can be achieved by loading molecular probes (integrin, stabilin-2, CD44, cRGD, ανβ3 peptide) on NPs [51]. Due to the advantages of NPs, some investigations of ROS-based NPs have been carried out to explore stronger and longer ROS scavenging effects for AS treatment. The main strategies of ROS-based NPs for AS treatment, which are organized in Table 1, have increasingly attracted the close attention of researchers in recent years.

### 4.1. Organic NPs

Organic NPs contain many advantages, including formulation biocompatibility, high bioavailability, simplicity, and the ability to carry large payloads [61,62]. Organic NPs are usually composed of small organic or polymer molecules. Lipid-based NPs, such as liposomes and lipid nanoparticles (LNPs), have a familiar spherical structure formed by a lipid bilayer [63]. Liposomes can form single and multilayer vesicle structures by rotary steaming. Different from liposomes, LNPs can form a micellar structure in the core of the particle; through this, the morphology can be changed according to formulas and arguments [64]. Similarly, polymer NPs, which can be synthesized from monomers or prefabricated polymers, are also common organic NPs because of their good biocompatibility [65]. Polymers usually include chitosan, poly lactic polyethylene glycol (PEG) and glycolic acid (PLGA) [66,67]. In addition, protein-based NPs prepared from collagen, keratin, soy protein or elastin, can also form polymeric particles.

#### 4.1.1. Organic NPs as Drug Carriers for ROS Scavenging

To date, most organic NPs carriers are used to deliver antioxidants (such as andrographolide, berberine and statins) [61,62,68]. Statins are widely orally administered cholesterol-lowering drugs that indirectly upregulate LDL receptor expression in liver cells. Due to the biotransformation in the liver, it is difficult for normal doses of statins to enter the systemic circulation, and increasing the dose of the drug is not feasible. Excessive amounts of drugs can cause damage to non-target organs. To solve this problem, Duivenvoorden et al. produced statin-loaded NPs constituted of high density lipoprotein cholesterol ([S]-rHDL), which can be intravenously injected, increase bioavailability, and improve efficiency in drug delivery to the plaque (Figure 1) [69]. The key finding of their study was that they were able to suppress the ROS production of macrophages through inhibiting the intracellular mevalonate pathway. [S]-rHDL accumulating in the plaque and being absorbed by macrophages significantly reduced inflammatory response and plaque volume. Finally, they confirmed that a 3-month low-dose [S]-rHDL treatment regimen could inhibit the progression of plaque inflammation, while a 1-week high-dose treatment regimen could significantly reduce the inflammatory progression of advanced atherosclerotic plaques. [S]-rHDL is a new, powerful AS nanotherapy that directly affects plaque inflammation. In addition, Ma et al. synthesized a functional nano-vector (BT_1500_M), which was formed by the esterification of VE succinate with PEG to encapsulate berberine (BBR, Figure 2) [68]. Their data showed that BT_1500_M increased BBR deposition in the liver and adipose by 107.6% and 172.3%, respectively. In in vitro and in vivo, the investigation found that BT_1500_M changed AMPK and NF-κB gene expression, successfully inhibited the triggering of macrophage activation and interrupted the cross-talk process between adipocytes and macrophages, which may contribute to its anti-inflammatory effect. Another study demonstrated that BT_1500_M reduced endothelial lesions, macrophage activation, cytokine release, and cholesterol ester accumulation in the aortic arch, thereby improving arterial plaque formation. This study provided a practical strategy for treating AS using a BBR-entrapped nanosystem.

#### 4.1.2. Organic Self-Assembled NPs as ROS Scavengers

In addition, some organic NPs themselves are capable of removing ROS. For instance, Wang et al. synthesized broad-spectrum ROS-scavenging NPs by covalently conjugating a SOD mimetic agent Tempol (Tpl) and a CAT mimic compound of phenylboronic acid pinacol ester (PBAP) onto a cyclic polysaccharide β-cyclodextrin (abbreviated as TPCD, Figure 3) [55,70,71]. TPCD passively targeted dysfunctional endothelial cells and migrated to inflammatory cells. Then, TPCD could be quickly and efficiently internalized by vascular smooth muscle cells (VSMCs) and macrophages. The desirable therapeutic outcomes of TPCD NPs were realized by reducing systemic and local oxidative stress and inflammation as well as by attenuating inflammatory cell infiltration in plaques. More importantly, the therapeutic strategy stabilized atherosclerotic plaques with lower numbers of macrophages and fewer, smaller necrotic cores and cholesterol crystals. In general, TPCD NPs were found to be a promising anti-AS nanotherapy worthy of further development. Moreover, ROS scavengers with properties capable of realizing the continuous and adjustable release of drugs have been developed [49,68]. For example, Chmielowski et al. synthesized a new class of NPs (1cM-PFAG), based on diglycolic acid (PFAG) loading with the core of ferulic acid (Figure 4) [49]. PFAG was found to lower ROS production by human monocyte derived macrophages (HMDMs), which were vital for promoting the growth of macrophages and preventing cell death. 1cM-PFAG had the highest bioactivity for limiting oxidized LDL (oxLDL) uptake, at an oxLDL concentration of 5 μg/mL. In addition, the 1cM-PFAG formulation was statistically significant (*p* < 0.05) for inhibiting oxLDL uptake compared with the control group. The researchers demonstrated that the optimal release rate of ferulic acid was associated with the diacetic acid ligand within the main chain of poly (anhydride), which minimized oxLDL uptake and ROS levels in HMDMs. In addition, the hydrophobicity of ferulic acid conjugates and the selection of chemical conjugates were key factors in the formulation of bioactive NPs. In general, loading drugs in the NPs of ROS-scavenger properties could play a synergistic treatment role, thereby significantly improving the therapeutic effect on AS.

### 4.2. Inorganic NPs

Inorganic NPs, which possess particular physical, chemical and magnetic properties, are uniquely endowed with certain applications, such as for imaging, diagnosis and therapy [51]. The advantages of inorganic NPs include their relative inertness, ease of synthesis and surface functionalization. However, the main disadvantages of inorganic NPs are their low solubility and potential toxicity, which limit their further clinical applications [72,73].

#### 4.2.1. Inorganic NPs as Drug Carriers for ROS Scavenging

In recent years, the inorganic nanocarriers used for the treatment of AS have mainly been metal nanomaterials. For example, Kim et al. explored loading iron oxide nanocarriers with interleukin 10 (IL-10) against AS [60]. IL-10 is an extensive anti-inflammatory cytokine used to reduce the production of ROS and regulate oxidative stress [74,75]. In their study, IL-10 was able to be easily encapsulated into the nanocarriers (NC), named IL-10-NC, and allowed sustained release under gentle conditions. Furthermore, compared with the free IL-10 and PBS groups, the IL-10-NC treated group indicated significant inhibitory effects on the progression of atherosclerotic plaques, as evident from the significant (*p* < 0.01) reduction in the relative blocked area (~22%, Figure 5). These findings suggested that targeting the delivery of anti-inflammatory cytokines to modulate the pro-inflammatory environment in plaques could be a promising strategy for the treatment of AS.

#### 4.2.2. Inorganic NPs as ROS Scavengers

Inorganic NPs with antioxidant effects, such as Mn, Ce, Cu, Se, V and noble metal, offer a novel ROS scavenging method. An early study reported that cerium oxide NPs could protect myocardial cells from oxidative stress [76]. Subsequently, Wu et al. developed new types of iron oxide–cerium oxide core–shell NPs (Fe_3_O_4_/CeO_2_) as possible therapeutic materials for ROS-related diseases (Figure 6) [77]. Fe_3_O_4_/CeO_2_ NPs were capable of reacting with ROS and of being detected by magnetic resonance imaging (MRI), showing enormous potential for diagnosis and treatment of AS. To date, Fe_3_O_4_ NPs have been extensively studied in cancer [78]. Moreover, inorganic NPs with antioxidant effects could be combined with traditional antioxidants. Atorvastatin combined with nano selenium (Nano-Se) has been found to significantly enhance the activity of serum GPx-1 and SOD and effectively avoid damage to oxidative stress caused by lipid metabolism disorders [79]. In recent years, inorganic NPs with natural enzyme properties have been found to effectively remove ROS, which has aroused widespread attention in the scientific community. A major breakthrough in this field also proves that some inorganic NPs can be endogenously metabolized with the help of biological ROS [50]. These strategies show great potential for the treatment of ROS-related inflammatory diseases.

### 4.3. Biomimetic NPs

As well as synthetic NPs, biomimetic NPs are also being explored for use in drug delivery systems due to their superior properties. In recent years, NPs coated with cell membranes, such as erythrocyte membranes [80], platelet membranes [74] and macrophage membranes [74], which have attracted much attention due to their excellent biocompatibility, can prolong NPs’ circulating half-life and protect them from being cleared by the reticuloendothelial system. The cell-membrane-coating drug delivery method may be more suitable for the treatment of inflammatory diseases than the living cell method. For instance, Gao et al. reported on biomimetic NPs coated with macrophage membranes [81]. These NPs were prepared via the self-assembly of amphiphilic chitosan oligosaccharide. Not only can macrophage membranes prevent the removal of NPs from the reticuloendothelial system, but they can also help NPs enter the inflamed tissue. At the same time, macrophage membranes were found to inhibit local inflammation by isolating pro-inflammatory cytokines. Moreover, Song et al. designed liposomes (P-Lipo) encapsulated by platelet membranes for targeting treatment of AS [82]. Due to its platelet membrane, P-Lipo was commendable for its long circulation properties and superior targeting. Furthermore, P-Lipo loading with Rapamycin (RAP) significantly reduced the growth of atherosclerotic lesions to 13.83 ± 2.09% (*p* < 0.001 when compared with any other group). With its potential homing ability and deeper penetrability of atherosclerotic plaques than conventional liposomes, P-Lipo could provide a safe and effective option for the treatment of atherosclerosis and many other platelet-related diseases.

### 4.4. ROS-Responsive NPs

With the growing demand for drug delivery to higher standards, ROS-responsive NPs have increasingly attracted the interest of researchers [67,83]. ROS-responsive NPs are developed to release drugs only at the target sites which generate excess ROS. Through this, these NPs could contribute to both improving treatment efficiency and reducing adverse side effects [53,54,67,83,84,85,86,87,88]. In consideration of the excess ROS level in an abnormal organization, many ROS-sensitive materials, including polythioether ketal, ketone mercaptan, ferrocene and selenium-containing copolymers, have been explored [52,57]. Hou et al. synthesized a new ROS-sensitive and CD44 receptor targeting amphiphilic carrier material, oligomeric hyaluronic acid-2′-[propane-2,2-diyllbls (thio)] diacetic acl-hydroxymethylferrocene (oHA-TKL-Fc), named HASF (Figure 7) [52]. Then, they combined curcumin (Cur) with HASF into nano-micelles (HASF@Cur micelles) by the self-assembling method. At an excessive ROS level, the thioketone bond of HASF could break. A release study in vitro showed that the efficiency of Cur release from HASF micelles was related to the concentration of ROS. Moreover, an in vivo study showed that the lesion area was reduced to 47.3 ± 3.4% when treated with HASF@Cur micelles, compared with the Cur group (63.2 ± 2.7%). HASF@Cur micelles had the more remarkable effect in reducing the lesion area. Dou et al. synthesized ROS-responsive NPs by chemical modification based on β-cyclodextrin [85]. β-cyclodextrin is a novel, ROS-sensitive material. In this strategy, they demonstrated that Rapamycin was released in an ROS-responsive manner. Furthermore, their study found that ROS-responsive NPs, which trigger almost no inflammatory response in in vitro or in vivo, effectively inhibited macrophage proliferation and foam cell formation through endocytosis and intracellular release of cargo molecules from macrophages. In addition, the sites overexpressing ROS can be used as targets for directed delivery of therapeutic agents and accurate fluorescence diagnosis. Furthermore, Ma et al. constructed therapeutic NPs (TPP) with continuous ROS responsiveness and two-photon AIE bioimaging for multimodal diagnosis and precise treatment of the lesions of excess ROS [53,84]. Prednisolone was connected to the two-photon fluorophore through ROS-sensitive bonds to form the diagnostic and therapeutic compound TPP, then loaded by the amphiphilic polymer self-assembled into a core–shell structured micelle (Figure 8) [53]. After the micellar interruption was triggered by excessive ROS, the TPP was able to react to the inflammatory environment and further release the drugs inside. As a result, the NPs showed a favorable ability of efficient AS treatment via effective anti-inflammatory activity and high resolution diagnosis of inflammation.

## 5. Challenges and Future Perspectives

AS is a chronic inflammatory disease that leads to the progressive narrowing and severe thrombosis of the coronary artery, cerebral artery and peripheral vessels. Although current lipid-lowering strategies are effective in slowing the progression of AS, the risk of cardiovascular events remains high. Furthermore, there are postoperative complications related to stent-assisted therapies or the coronary artery, such as vascular inflammation and re-stenosis. Since oxidative stress plays a key role in the inflammatory course of AS, a growing number of literature studies have focused on developing antioxidant strategies targeting ROS to prevent and treat AS lesions.

Through the exploration of more advanced nanomaterial designs, ROS-based NPs have been found to not only improve the ability to target lesions actively or passively, but also to be able to be released in response to demand. These NPs effectively inhibit the formation of foam cells and significantly improve the stability of atherosclerotic plaques. More importantly, based on these strategies, we may realize tailor-made treatments according to the individual in order to minimize unnecessary toxicity. Consequently, ROS-based NPs are promising and deserve further development. However, it should be noted that human physiology is a complex system. Non-targeted areas also have similar oxidative stress levels. These NPs that respond to oxidative stress may be released in areas other than the diseased area. In summary, ROS-based NPs for the treatment of AS are in the exploratory stage. The following problems are still worth paying attention to. Firstly, there is a dynamic equilibrium between NP activity and intracellular levels that is influenced by the ROS-scavenging capability and action time. Sencondly, the location and concentration of ROS in the body should be monitored in real time and effectively assessed. Thirdly, the complexity of atherosclerotic pathogenesis and the lack of innate targeting mechanisms significantly increases the difficulty of effective treatment. Last but not least, the safety of ROS-based NPs should be fully guaranteed. Owing to the latest advances in biomedicine and nanotechnology, researchers can develop more accurate and systematic treatment strategies to improve their results, which should be based on the elucidation of clinical genes and clinical symptom regulation mechanisms.

## Figures and Tables

**Figure 1 materials-14-06921-f001:**
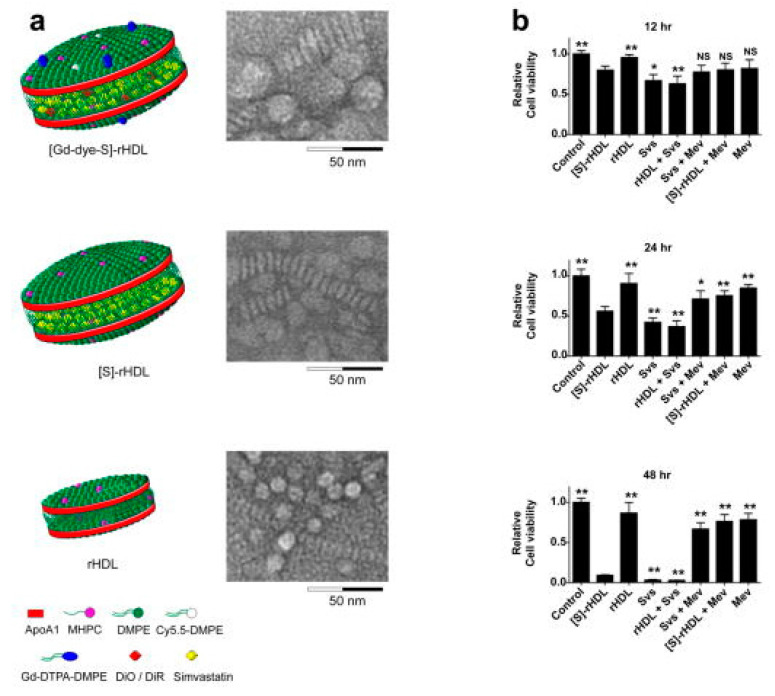
Schematic representations of nanoparticle formulations and in vitro efficacy data. (**a**) Schematic representation of dual gadolinium and fluorescent dye (Cy5.5, DiO, DiR) labeled statin containing reconstituted high density lipoprotein ([Gd-dye-S]-rHDL), statin containing rHDL ([S]-rHDL), and rHDL. (**b**) In vitro cell viability assays of murine macrophages (J774A.1), incubated with combinations of [S]-rHDL (10 μM statin) free simvastatin (10 μM), rHDL plus free statin (10 μM), free statin (10 μM) plus mevalonate (100 μM), [S]-rHDL (10 μM) plus mevalonate (100 μM), and only mevalonate (100 μM). There was also a control group of cells not incubated with anything. Macrophage cell viability was markedly decreased in the [S]-rHDL and free statin group. This effect was abolished by the addition of mevalonate, indicating that the effect of HMGR inhibition on cell viability is mediated through the mevalonate pathway. N = 6 for all bars. * *p* < 0.05, ** *p* < 0.01. Reprinted from Nature communications. Duivenvoorden, R., et al., A statin-loaded reconstituted high-density lipoprotein nanoparticle inhibits atherosclerotic plaque inflammation, pages 3065. Copyright 2014 [69].

**Figure 2 materials-14-06921-f002:**
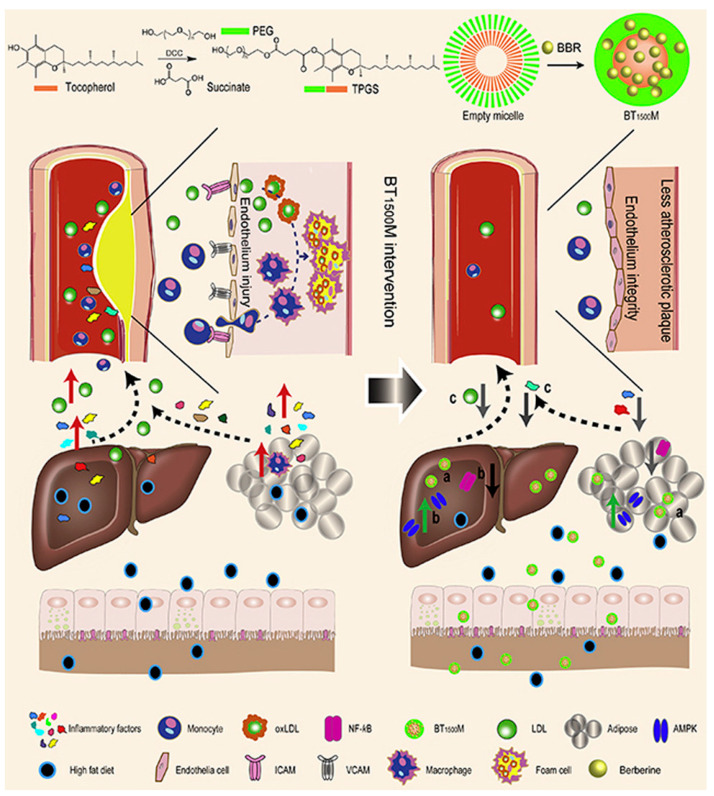
These NPs, which increased gut absorption and intra-cellular uptake of berberine, modulated AMPK and NF-κB expression, and improved dyslipidemia and inflammation induced by a high fat diet. Endothelial injury and subsequent macrophage infiltration and cholesteryl ester gathering in the aortic arch were decreased, resulting in the inhibition of artery plaque build-up. Reprinted from Acta Pharmaceutica Sinica B. Ma, X., et al., Functional nano-vector boost anti-atherosclerosis efficacy of berberine in ApoE^−/−^ mice, pages 1769–1783. Copyright 2020, with permission from Elsevier [68].

**Figure 3 materials-14-06921-f003:**
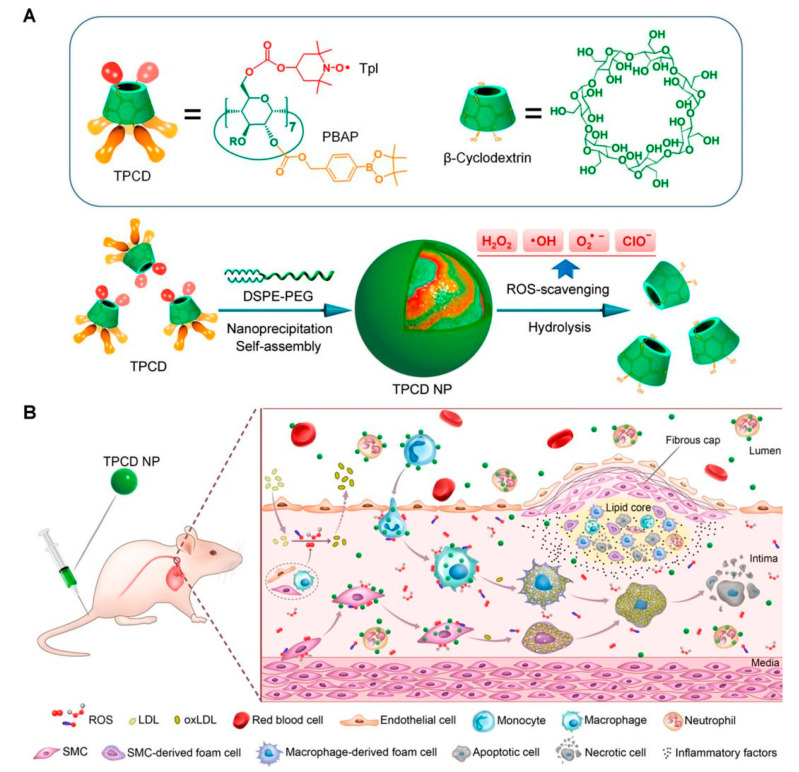
Schematic of engineering a broad-spectrum ROS-scavenging nanoparticle and targeted therapy of AS. (**A**) Chemical structure of a broad-spectrum ROS-eliminating material TPCD and development of a TPCD nanoparticle (TPCD NP). (**B**) Sketch of targeted treatment of AS by eliminating ROS through i.v. administration of engineered TPCD NP. Reprinted with permission from ACS nano. Wang, Y., et al., Targeted Therapy of Atherosclerosis by a Broad-Spectrum Reactive Oxygen Species, pages 8943–8960. Copyright 2018 American Chemical Society [55].

**Figure 4 materials-14-06921-f004:**
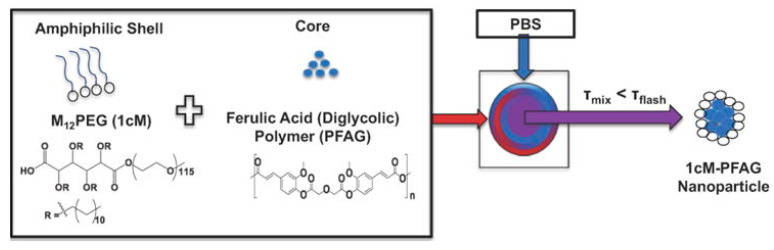
Description of the flash nanoprecipitation process for the formulation of the nanoparticles. Reprinted with permission from Acta biomaterialia. Chmielowski, R.A., et al., Athero-Inflammatory Nanotherapeutics: Ferulic Acid-based Poly(anhydride-ester) Nanoparticles Attenuate Foam Cell Formation by Regulating Macrophage Lipogenesis and Reactive Oxygen Species Generation, pages 85–94. Copyright 2018 [49].

**Figure 5 materials-14-06921-f005:**
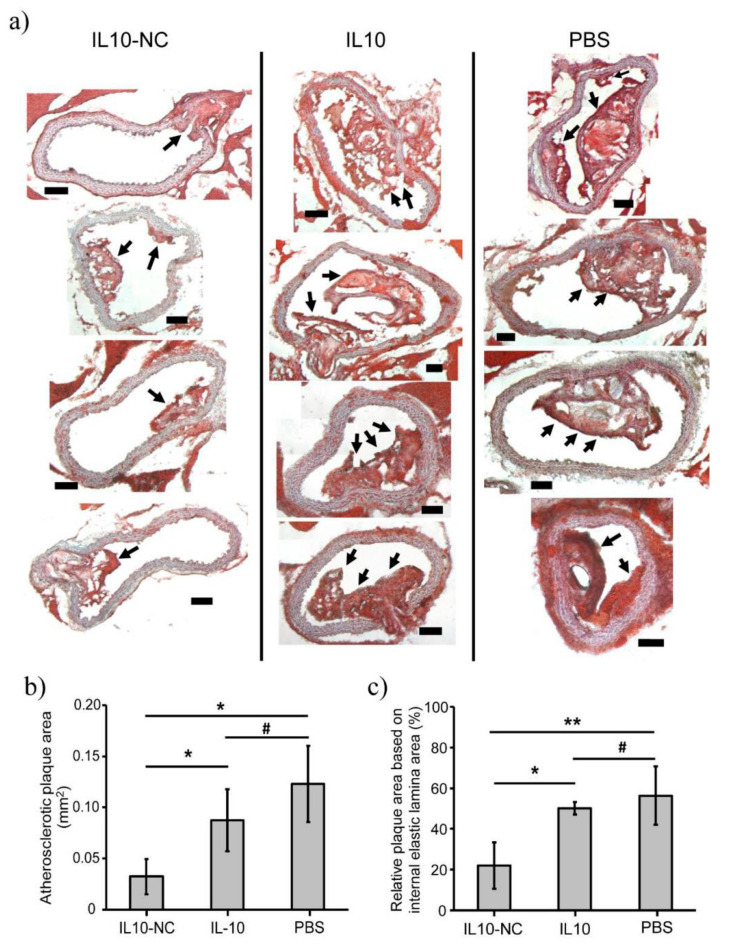
(**a**) Oil red O (ORO) staining of plaques in aortic root cross-sections after i.v. injection of IL-10-NC, IL-10, and PBS. Samples were injected once a week for 3 weeks. Black arrows indicate plaques. Quantitative analysis of relative atherosclerotic plaque area (**b**) from the ORO-stained (**a**) images, and (**c**) based on internal elastic lamina area and atherosclerotic plaque area (*n* = 4, * *p* < 0.05, ** *p* < 0.01, and # not significant). Reprinted from Biomaterials. Kim, M., et al., Targeted delivery of anti-inflammatory cytokine by nanocarrier reduces atherosclerosis in ApoE^−/−^ mice, pages 119550. Copyright 2020, with permission from Elsevier [60].

**Figure 6 materials-14-06921-f006:**
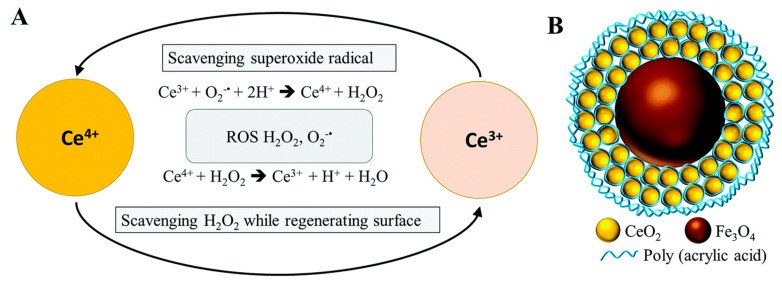
Regenerative antioxidant properties of cerium oxide and the simulation structure of iron oxide/cerium oxide core-shell nanoparticles. (**A**) Regenerative antioxidant properties of cerium oxide show that cerium ions are capable of scavenging superoxide radicals when trivalent cerium ions change to tetravalent cerium ions. On the other hand, cerium oxide has the ability to scavenge H_2_O_2_ when tetravalent cerium ions become trivalent cerium ions. (**B**) Simulation structure of iron oxide/cerium oxide core-shell nanoparticles. Reprinted with permission from Journal of Materials Chemistry B. Wu, Y., et al., Novel iron oxide-cerium oxide core-shell nanoparticles as a potential theranostic material for ROS related inflammatory diseases, pages 4937–4951. Copyright 2018 Royal Society of Chemistry [77].

**Figure 7 materials-14-06921-f007:**
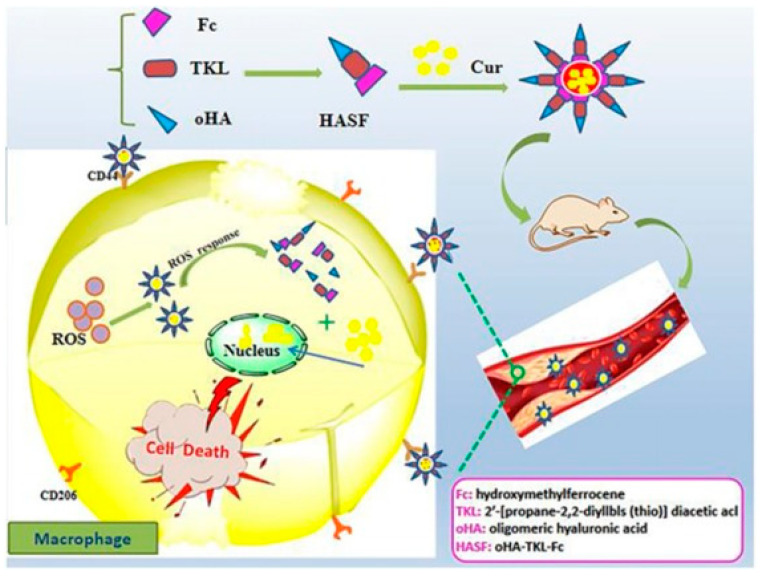
Cur release mechanism diagram of dual ROS-sensitive and CD44 receptor targeting nanoparticles. Reprinted with permission from Carbohydrate polymers Hou, X., et al., Novel dual ROS-sensitive and CD44 receptor targeting nanomicelles based on oligomeric hyaluronic acid for the efficient therapy of atherosclerosis, pages 115787. Copyright 2020, with permission from Elsevier [52].

**Figure 8 materials-14-06921-f008:**
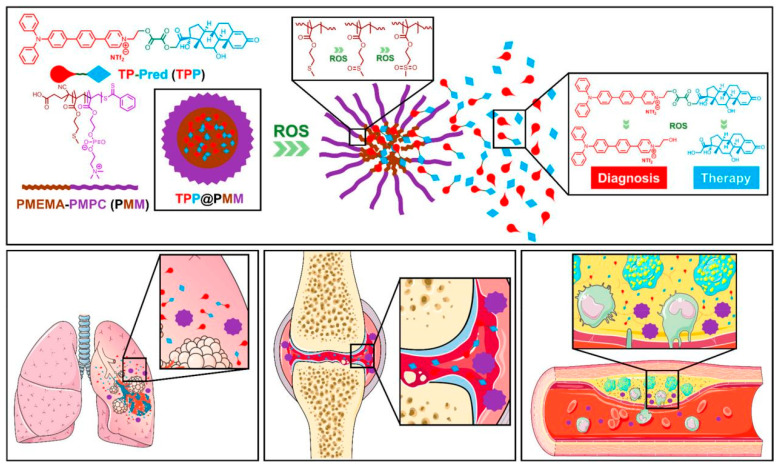
Illustration of the NPs with two-photon imaging and serial ROS sensitivity. Reprinted with permission from ACS nano. Ma, B., et al., Reactive Oxygen Species Responsive Theranostic Nanoplatform for for Two-Photon Aggregation-Induced Emission Imaging and Therapy of Acute and Chronic Inflammation, pages 5862–5873. Copyright 2020 American Chemical Society [53].

**Table 1 materials-14-06921-t001:** ROS-based NPs.

Model (In Vitro)	Model (In Vivo)	NP Composition	Size (nm)	Drug Load	Ref.
Raw 264.7	High-fat-fed rats	Oligomeric hyaluronic acid	150.8	Curcumin	[52]
Raw 264.7	ApoE^−/−^ (male) mice	β-cyclodextrin	57.5	Prednisolone	[53]
Human monocyte-derived macrophages	N/A	PFAG	160–300	Ferulic acid	[49]
Human umbilical vein endothelial cells	N/A	PVAX	200	Vanillin	[54]
Raw 264.7	ApoE^−/−^ (male) mice	β-cyclodextrin	128	Tempol and phenylborate	[55]
Raw 264.7	C57BL/6 mice fed with high-fat diet for 10 weeks	Indole	300	Propofol	[56]
Raw 264.7	Rats fed with high-fat diet for 2 weeks	Hyaluronic acid	69.9	Dexamethasone and MTOR	[57]
Vessel smooth muscle cell	Rats fed with high-fat diet for 10-12 weeks	Amino acid	200	NOX	[58]
Human umbilical vein endothelial cells	ApoE^−/−^ mice fed with high-fat diet for 16 weeks	PLGA	160	Atorvastatin calcium, LOX1	[59]
Macrophage cells	ApoE^−/−^ mice	Iron oxide	80	IL-10	[60]

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
