# Peer review of "ROS-Based Nanoparticles for Atherosclerosis Treatment"

_materials, 2021, doi:10.3390/ma14226921_

Round 1

Reviewer 1 Report

In the manuscript [materials-1414266] entitled “ROS-Based Nanoparticles for Atherosclerosis Treating” the authors reviewed literature data on atherosclerosis, including the relationship between reactive oxygen species and atherosclerosis, strategies to reduce reactive oxygen species for atherosclerosis treating, and reactive oxygen species based nanoparticles for atherosclerosis treating (detailing several types of nanoparticles of interest in this respect). The data presented and discussed are very interesting, and the topic of the manuscript is relevant to the field of the Materials journal. The paper has a good structure, and is based on adequate and mostly recent references. The manuscript is in general well written (apart from a number of ambiguous formulations – mentioned below), and is comprehensive but still concise. However, there are 5 minor improvements to be performed by the authors before recommending the publication of the paper.

A. The authors have to add at the end of Introduction section a short paragraph describing the methodology used: what databases were used to search for finding the articles of interest, the key words used, the preliminary obtained results (number of articles found), criteria for including the found papers into their review (or for excluding other papers), etc.

B. The authors have to recheck almost the entire paper for the correct order of citing the references into the text, for four reasons. Firstly, refs [2] and [17] in the References list are identical. After deleting [17] from the list, the numbering order will be changed. Secondly, in p.3 line 6 in the first paragraph of subsection “3. Strategies to Reduce ROS for Atherosclerosis Treating” is cited ref. [34], and the next one cited, 9 lines below, is ref. [37]. In these circumstances, the authors either will add the missed references [35], and [36] (taking into account the new numbering after correction of the previous issue) somewhere in the space between [34] and [37], or they will change again the number of all cited references, after ref. [34]. Thirdly, and in the same context, the authors arrived to ref. [81] immediately after ref. [46] p.3 line 3 in the second paragraph of subsection “3. Strategies to Reduce ROS for Atherosclerosis Treating”. Finally, ref. [62] was improperly used in p. 4 line 2 of paragraph 4.1.

C. The authors should detail the legends of the used figures. Also, they have to mention “used with permission” at the end of each legend. The second part of Figure 3 does not have relevance for this review.

D. The authors have to recheck the entire manuscript for grammar errors and ambiguous phrases that have to be reformulated:

1) please replace “incubation” with “progress” in Abstract, line 2.

2) in Abstract, please use the same tense (in lines 6-7 is Present Perfect, and in line 9 is Past Tense).

3) in p.2 (Introduction), in line 6 ROS is an acronym that defines a plural form, and next, in line 7, ROS is considered as singular thus the accord subject-predicate has to be corrected.

4) in p.2 (Introduction), line 7 has to be reformulated or “demonstrates” replaced with “demonstrating”.

5) in p.2 (Introduction), lines 9, 15 – the same comment as in 3).

6) in p.2 (Introduction), the last 5 lines 9, 15 – the same comment as in 2).

7) in p.2 (paragraph 2.1), line 3, please delete “of” after “especially”.

8) in p.2 (paragraph 2.1), line 11, please add “of” after “excess”.

9) in p.3 line 1, the same comment as in 2).

10) in p. 3 line 3 is explain the acronym DNA, but no others that follow, such as MAPK, JAK/STAT, etc.

11) please reformulate lines 6,7 in p. 3, since collagen is not a cell, and cytoplasm is not an organelle.

12) in p.3 line 3 of subsection 3, cytochrome-oxidase belongs to the inner mitochondrial membrane, it is not a “submembrane complex”. Please reformulate.

13) in p.3 please reformulate lines 5, 8 and 15-16 of subsection 3.

14) in p.3 please reformulate line 2 of subsection 4.

15) in p.4 Table 1, “Prednisoloe” is not correct. Maybe “Prednisolone”.

16) in p.4 line 8 of paragraph 4.1 please use Present Tense instead of Present Tense Continuous.

17) in p.4 line 2 of paragraph 4.2 please add “indirect” before “upregulating”.

18) in p.5 please reformulate lines 2 and 4.

19) in p.6 lines 6 and 8 please reformulate for correct subjects-verbs agreements (NPs was defined as plural, and mentioned here in line 5 “These”); the same comment for line 10 – it should be “cores”.

20) please reformulate last line of the paragraph in p. 6.

21) in p. 7 please reformulate lines 1-3 and 6 of the paragraph 4.2.1.

22) in p. 7 please correct “encapsulationa” in line 4

23) in p. 7 please reformulate lines 4-5 of the paragraph 4.2.2.

24) please reformulate the last line in p.7 and the first line in p.8.

25) please reformulate line 2 in p.8.

26) please reformulate lines 6, 11 and 14 in p.8, paragraph 4.3.

27) in p.9 line 1 please use Past Tense (synthesized).

28) in p.9 line 2 please explain HASF.

29) in p.9 line 3 please replace “could breakage” with “could break”.

30) in p.9 line 4, the correct Past Tense form is “led” not “leaded”.

31) in p.9 line 6, please use Past Tense (depended), the same comment for line 9 (triggered).

32) in p.9 lines 13 and 14 please be consistent in using either “twophoton” or “two-photon”

33) please reformulate the last 3 lines in p.9.

34) please reformulate lines 1-3 and the last 3 lines in the second paragraph of subsection 5.

E. The authors have to be consistent with the formatting of references in the References list. In almost all references the authors have to format correctly the title of the papers, or the journal’s name, in two cases [12] and [22] the references are not complete, and ref. [89] does not look as a reference.

Reviewer 2 Report

The paper discuss about an interesting topic with a clear arrangement of the sections. Overall, the concepts are quite general. When talking about the effect of a treatment or similar it would be very useful to provide the reader with more details such as a quantification, how it was determined and in which conditions. I suggest to not only state the effect of the treatment but also describe critically the experimental details that of course determine the results. All the paragraph needs to be improved with specific experimental, quantitative details critically presented and compared to each other. I suggest acceptance after major revision

English has to be improved.

  • More details about the current state of the clinical treatments for AS would be highly appreciated.
  • I would suggest to improve the language and correct typos, such as “et al.”. Rewrite the last sentence of Paragraph 4. P
  • Paragraph 4.1.2: The first two lines need to be rewritten. The last two line are confusing.
  • Words such as “efficient”, “fast”, “lower”, “extensive”, “effectively”, “good” and similar need to be contextualized and explained in the entire text. I suggest to report data that can quantify the efficiency and the rapidity, experiments which can explain how the effect was evaluated in order to allow the reader to make a more complete and rational evaluation of the different aspects. The reader need this tools for comparing the various treatments.
  • Usage of past or present tenses should be carefully checked and it should be coherent in the entire text.
  • Paragraph 4.4: “Hou et al synthetise a new type of targeting carrier material with dual ROS-sensitive and CD44 receptors, named as HASF, to realize high encapsulation of curcumin (Figure 5)[50]. In the plaque of AS, the thioketone bond of HASF, which changed from hydrophobic to hydrophilic, could breakage at excessive ROS level. So, the NPs leaded to the release of curcumin when destroyed, which increased drug penetration and anti-inflammatory effect. A release study in vitro further showed that the curcumin released from HASF micelles depends on the ROS condition.” This paragraph needs to be rewritten and the English needs to be checked, it is not clear.
  • Paragraph 4.4 needs a grammar check. Sentences are too complex and not clear.
  • It would be appreciable a critical overview discussing the advantages and the disadvantages of the ROS approach compared to other approaches.

Round 2

Reviewer 2 Report

The manuscript has been sufficiently improved and it is ready for publication.